# Soil *Aspergillus* Species, Pathogenicity and Control Perspectives

**DOI:** 10.3390/jof9070766

**Published:** 2023-07-20

**Authors:** Queenta Ngum Nji, Olubukola Oluranti Babalola, Mulunda Mwanza

**Affiliations:** 1Food Security and Safety Focus Area, Faculty of Natural and Agricultural Sciences, North-West University, Private Bag X2046, Mmabatho 2735, South Africa; queenbrighten@yahoo.com (Q.N.N.); olubukola.babalola@nwu.ac.za (O.O.B.); 2Department of Animal Health, Faculty of Natural and Agricultural Sciences, North-West University, Private Bag X2046, Mmabatho 2735, South Africa

**Keywords:** *Aspergillus* species, ecology, pathogenicity, soil, biological control

## Abstract

Five *Aspergillus* sections have members that are established agricultural pests and producers of different metabolites, threatening global food safety. Most of these pathogenic *Aspergillus* species have been isolated from almost all major biomes. The soil remains the primary habitat for most of these cryptic fungi. This review explored some of the ecological attributes that have contributed immensely to the success of the pathogenicity of some members of the genus *Aspergillus* over time. Hence, the virulence factors of the genus *Aspergillus*, their ecology and others were reviewed. Furthermore, some biological control techniques were recommended. Pathogenic effects of *Aspergillus* species are entirely accidental; therefore, the virulence evolution prediction model in such species becomes a challenge, unlike their obligate parasite counterparts. In all, differences in virulence among organisms involved both conserved and species-specific genetic factors. If the impacts of climate change continue, new cryptic *Aspergillus* species will emerge and mycotoxin contamination risks will increase in all ecosystems, as these species can metabolically adjust to nutritional and biophysical challenges. As most of their gene clusters are silent, fungi continue to be a source of underexplored bioactive compounds. The World Soil Charter recognizes the relevance of soil biodiversity in supporting healthy soil functions. The question of how a balance may be struck between supporting healthy soil biodiversity and the control of toxic fungi species in the field to ensure food security is therefore pertinent. Numerous advanced strategies and biocontrol methods so far remain the most environmentally sustainable solution to the control of toxigenic fungi in the field.

## 1. Introduction

Filamentous fungi of the *Aspergillus* genus are cosmopolitan, saprophytic, asymptomatic endophytes and opportunistic phytopathogens. The number of *Aspergillus* species is reported to be between 300 and 400 [1]; arguably, this number will continue to rise as new species are described. Five *Aspergillus* sections (Fumigati, Flavi, Nigri, Terrei, and Nid ulante) have been reported to cause disease in humans among the 17 assigned sections in the family Aspergillaceae [2], and are therefore of health and economic relevance. Being the known opportunistic pathogens that they are, some species of the genus have been isolated from a wide variety of substrata of major biomes, including soil and litter [3,4,5], and are main players in the degradation processes of organic matter in ecosystems. A thorough understanding of the environmental strains’ evolutionary dynamics leading to their pathogenicity and of the interactions with organic and inorganic diversity of these diseased-causing species will ensure that effective control measures will be taken. Mead, Steenwyk [6] revealed that the evolution of *Aspergillus* pathogenicity involved both conserved and species-specific genetic contributors. *Aspergillus* species since their discovery in the 1960s have been studied for their ability to produce mycotoxins that contaminate agricultural products and that are a threat to the health of humans and animals and by consequence a threat to food security; for instance, from 60–80% of global food crops are contaminated by mycotoxins [7]. These high percentages of mycotoxins of crops has been estimated to cause losses in the billions of dollars in the United States of America [8] and also an estimated loss of USD 17.28 million due to mycotoxin contamination in the Kenyan dairy industry [9]. Furthermore, some of the most severe public health issues caused by mycotoxins since the 1960s, when they first became of concern, have been extensively studied. The most tragic human implication of aflatoxin contamination of food recorded in history, which occurred in Kenya where 125 deaths were recorded in 2005, cannot be forgotten easily [10]. The real implication of these cryptic species, together with species previously considered not to be of clinical concern, could be that they synthesize metabolites apart from common mycotoxins that might be underestimated contributors to toxicity in humans and animals [11,12]. It is critical to study members of the *Aspergillus* genus in detail, especially the disease-causing species, including their origin, ecology, and their pathogenicity to be able to effectively mitigate against their effects. The biodiversity of the representative species from the five *Aspergillus* sections known to cause diseases in humans will be explored in this review in terms of their ecological dominance, pathogenicity potential, and possible control techniques.

## 2. Ecology

Details of *Aspergillus* species ecology and cell biology which involves metabolic adjustment to nutritional and biophysical challenges contribute to their status as arguably the most potent opportunistic fungal pathogens of mammalian hosts. Strains of *A. fumigatus*, *A. flavus*, *A. niger* and other *Aspergillus* species can inhabit different types of environments. These habitats vary widely with respect to biotic and abiotic factors. Even though *Aspergillus* species have been isolated from diverse environments, the soil remains their primary reservoir. Most of these *Aspergillus* species have received limited research regarding their role in soil fungal ecology. Soil is the natural habitat for toxin-producing fungi of *Aspergillus* species, for example *A. flavus* and black aspergilli, giving it the environmental advantage of mycotoxin production [13]. *Aspergillus* species populations are very diverse and their stability in the soil and on plants is ambiguous. The impacts of climate change on the emergence of new *Aspergillus* species and mycotoxin production are uncertain, and if climate change continues at its current pace, mycotoxin contamination risks in soil ecosystems will increase or decrease depending on the region under consideration [14,15]. *Aspergillus* species occur most frequently in regions with tropical and subtropical climates, most commonly found between 25–35 degrees latitude [16,17]. Atukwase et al. [18] has long established that high-altitude regions registered the highest total mycotoxin contamination in crops, whereas crops cultivated at mid- and low altitudes had less mycotoxin contamination. Additionally, rare and new species of *Aspergillus* have been reported frequently from tropical and subtropical soils [19]. The high adaptability of *Aspergillus* species to different environments allows for their survival at different temperatures, low water presence, and variations in soil pH and oxygen concentrations [20]. Interactions of *Aspergillus* species with biotic and abiotic components will be a deciding factor for the kind and role of secondary metabolites, such as aflatoxins, gliotoxin, patulin, cyclopiazonic acid, and ochratoxin, that will be produced [11].

Advances in environmental DNA analyses have greatly enhanced knowledge in the area of soil biodiversity distribution [21,22]. Toxigenic fungi show wide variations in their growth requirements and mycotoxin production. Limited work has been done on soil ecotoxicology to determine the fate of some fungal metabolites present in the soil environment [23,24], such as jasmonic acid, a metabolite synthesized as a defense response to *Aspergillus* species infection [25]. Hence, the consequences of some of these metabolites and their degradation products for soil biodiversity remain vague [23,26]. Environmental stressors like elevated temperatures, increased CO_2_ concentrations, extended drought, and rainfall variation directly affect the occurrence of *Aspergillus* species, favoring their growth, conidiation, and spore dispersal [27]. Moisture and temperature fluctuation patterns severely impact the development and geographical distribution of secondary metabolites produced by toxigenic fungi [14,28]. Soil temperature and moisture highly affect soil microbial activity, including the growth and extensive spread of mycotoxigenic fungi, thereby modifying host resistance and pathogenic interaction [29]. Gallo et al. [20] reported that temperature plays an important role in gene expression and the secretion of secondary metabolites. In drought-heated soil, increased total mycotoxin concentrations were reported in cultivated crops [15,26,30].

The organic content of soil and soil texture is of significance since mycotoxins strongly sorb to soil organic carbon and clay minerals [31,32,33]. Soil texture depends on the relative proportion of various sizes and particles of its constituents (sand, silt, and clay). Silty clay loam soil has a 50% adsorption ability, which prevents aflatoxins and their metabolites from leaching into groundwater [34]. Soils with higher organic matter and clay content have a greater moisture-holding ability, increasing the chances of survival of *Aspergillus* species’ propagules. Aflatoxins produced by these *Aspergillus* species associate with soil-binding sites, thereby resisting microbial degradation and causing extended aflatoxin contamination, surviving for up to 120 days [35]. Soil density impacts the prevalence of *Aspergillus* species; for example, high-density peanut-growing soils had the highest numbers of *Aspergillus* species as compared to lower soil densities [36]. Achaglinkame et al. [37] reported that light, sandy-type soils impacted the incidence and growth of *A. flavus* and increased aflatoxin contamination. Hence, soil type provides guidance regarding what to expect in terms of fungal biodiversity. Reddy et al. [38] stated that *Aspergillus* species are highly associated with highly productive soils.

Soil pH impacts fungal diversity, irrespective of the effects of latitude, climate, and historical effects that prevail in global-scale studies [39,40,41]. Ectomycorrhizal fungi and evergreen plants acidify soil by exuding organic acids and shedding recalcitrant litter [22,42]. Mira et al. [43] reported that low pH irreversibly damages the plasma membrane, inducing conformational changes in membrane proteins and causing leakage of ions and metabolites. Fungal populations were common in cultivable land with a low or moderate cation exchange capacity and organic matter content [44]. Movement of cations across the plasma membrane has been reported to be rapid and some *Aspergillus* species have been able to adapt to pH-induced stresses, as the cytosol in the plasma membrane acts as an osmotic barrier to resist extreme pH values. Even though most members of the genus *Aspergillus* survive optimally under slightly acidic conditions (pH 5.0–6.0), some species are tolerable to much lower pH (1.5) [45,46]. Increasing acidity makes the organic matter more soluble and vulnerable to leaching. Calcium concentration in soil affects the soil pH, influencing the richness and composition of fungal groups [47]. Activities of particular tree species in combination with a specific bedrock might lead to extreme pH values that result in impacts on soil fungi biodiversity [48,49]. Therefore, the exposure of soil organisms to trace elements is influenced by various mechanisms such as adsorption and release from soil-binding sites, interactions with the soil microbial community, and the metabolic transformations of toxins in the soil solution [50].

Fungi are primary decomposers of organic material, key root mycorrhizal symbionts of plants, and pathogens in the soil. *Flavi* isolates are normally saprophytic, polyketide metabolites which can increase fungal survival in soil. Such a benefit may be unnecessary in carbon-rich agricultural environments. Therefore, understanding the soil and the processes associated with the soil is crucial in developing sustainable agricultural systems. The loss of soil biodiversity affects soil quality, particularly in arable soils under intensive agriculture, and is therefore an unignorable serious global issue. Numerous soil organisms are endangered by shifts in land use, changing climate, and ecosystem management [51,52,53]. The recommendations by the World Soil Charter [54] emphasized the significance of soil biodiversity in supporting soil functions and regulating and maintaining various healthy ecosystem services. Tančić-Živanov et al. [44] analysed fungal biodiversity from different types of soils and identified *Aspergillus* species as one of the most frequently occurring genera. Increased populations of *Flavi* isolates in no-tillage or nonagricultural fields as compared to nearby tilled soils have been reported [55,56]. This is due to the soil being the main reservoir of global biodiversity. The soil is of significance in ecosystem services and is resilient to disturbances, with the ability to mitigate stress caused by global climate change [57]. Economic growth and human well-being, therefore, depend on healthy soils. Some soil changes, like erosion and metal pollution, are a result of human activities and are irreversible. Others could be reversible, like the decreases in organic matter. Fertilizers are used to provide nutrients, allowing for increased production without impoverishing the nutrient status of the soil. It is currently difficult to determine the extent to which soil contamination with fungal secondary metabolites may endanger soil health, leading to increased risks regarding the quality of cultivated crops. Maintenance of soil fertility is required for the development of sustainable agricultural food systems and should be intentional in a rapidly growing population in developing countries. The probability of systemic soil fungal infection is increased in the subsistence farming system by farmers who cultivate their fields with contaminated seeds from preceding harvests [58]. In irrigated soil, the incidence of fungi in plants is moderate, with no proof of mycotoxin secretion [59]. The removal of vegetation exposes soil to erosion and in wet regions increases leaching, resulting in acidic soils. Monoculture and late planting have been reported to increase fungal inoculum and pest damage leading to high fungal infection and high mycotoxin production in soils of agricultural produce [60]. Crop rotation and intercropping minimise soil mycotoxin contamination by breaking the infection cycle. For instance, the rotation of legumes like cowpea and soybean with maize can help break pest and disease cycles and improve soil fertility [61,62]. Gebreselassie et al. [63] reported a 68% reduction in levels of *Aspergillus* infection in fields where supplementary irrigation and tied ridge combination were practised. Also, soil moisture and fertility amendment practices reduce *A. flavus* infection levels [63].

## 3. Pathogenicity

A diverse set of mechanisms have driven the evolution of fungal pathogenicity and virulence over time and have profound effects on both host and parasite. These are influenced by environmental factors and genotypes [64]. For instance, the production of some toxic secondary metabolites is a survival mechanism under stressful conditions. The ability to express virulence factors under different conditions distinguishes pathogenic from non-pathogenic strains. While some microbes cause disease in the normal host, a larger number of microbes can cause disease in immunocompromised hosts; hence, *Aspergilli* are still counted among opportunistic pathogens in humans. Thus, many *Aspergillus* species can become pathogenic when the opportunity presents itself. Also, the degree of pathogenicity differs from one *Aspergillus* species to another. For instance, 100-fold doses of *A. flavus* have higher infectivity compared to *A. fumigatus* [65]. However, not all *A. flavus* strains are pathogenic. Some *A. flavus* strains incapable of producing aflatoxins have been called *A. oryzae*. Such isolates are routinely found in agricultural fields, with only some being classified as *A. oryzae.* The pertinent question here is: why are not all *A. flavus* incapable of producing aflatoxins classified as *A. oryzae*? Although *A. oryzae* is genetically very close to *A. flavus*, which is known to produce the most potent natural carcinogen, aflatoxin, *A. oryzae* has no record of producing aflatoxin or any other carcinogenic metabolites. This is because *A. oryzae* is said to be the domesticated species and, therefore, may be found only in a domesticated form but not in nature [66]. Another example of non-aflatoxigenic species is *A. sojae*, which is genetically related to *A. parasiticus*, an aflatoxin producer. These aflatoxigenic strains have been distinguished from the non-aflatoxigenic strains on the basis of their morphological and physiological differences, as well as toxicity [66,67]. *A. oryzae* strains, for instance, contain all or parts of the aflatoxin biosynthetic gene cluster, although they are non-aflatoxigenic. This is because homologs of aflatoxin biosynthesis gene cluster are not often expressed in *A. oryzae* even under conditions that are favourable to aflatoxin expression in *A. flavus.* For instance, the expressed sequence tag analysis showed a striking contrast in expression of the aflatoxin biosynthesis gene homologs; while in *A. flavus* all the 25 gene homologs were found, no such genes were found in *A. oryzae* except for *aflJ* and *norA* [68].

It has been predicted that virulent parasites will be more competitive during mixed infection; for instance, the rate of sporulation is increased in *A. flavus* in the presence of competitors, though not limited to co-infections with less virulent parasites [65]. Ehrlich et al. [56] reported no genetic exchange among *A. flavus* atoxigenic isolates and toxin-producing isolates. There has been a remarkable discovery of sexual stages amongst members of the genus *Aspergillus*, which were formerly assumed to be asexual [69,70,71]. The possible effects of sexual reproduction on fungal virulence include the ability of fungi to produce mycotoxins. This has been linked to the presence of specific gene clusters, where sexual recombination has explained this variation in mycotoxin production amongst members of this genus [70]. On the other hand, asexual sporulation and mycotoxin production in *Aspergillus* species is a survival mechanism induced in response to stressful conditions [72,73]. Fungal genomic analyses have shown that most secondary metabolite-associated gene clusters are silent, implying that fungi continue to be a source of underexplored bioactive compounds. For instance, Frisvad et al. [74] reported that aflatoxins are produced by almost 16 species. *Aspergillus flavus* possesses 56 secondary metabolite biosynthesis gene clusters, but only a few have been assigned to individual gene clusters [11,75]. Hence, other members of the genus *Aspergillus* might produce other metabolites apart from the well-known mycotoxins that could be underrated contributors toward its toxicity. Studies have predicted that parasites evolve towards an optimal level of virulence. This prediction was solely true for obligate parasites and host-parasite coevolution within closed systems, yet, the evolution of virulence in opportunistic parasites poses a challenge to standard virulence evolution predictions [76]. Therefore, transmission dynamics associated with how virulence evolves both within a host and between hosts are unpredictable due to epidemiological feedback [77]. Most *Aspergillus* infections are likely to be acute events, characterized by transient dynamics rather than parasite adaptation, which explains why ubiquitous and potentially virulent fungi have been so rarely recorded [65]. Pathogenicity of obligate parasites is maintained by natural selection and will increase or decrease as an evolutionary response to environmental conditions or transmission opportunity in the host population [78]. Rokas et al. [79] explained that the ability to cause disease in humans has evolved multiple times independently within *Aspergillus.* They used two models to explain the evolution of the pathogenicity within this genus, including species-specific and conserved pathogenicity models. In the first model, genetic determinants of virulence were shown to be unique to each pathogenic species. For example, *A*. *fumigatus* strains (Af293 and A1163) were compared against their closest non-pathogenic relatives (*A*. *fischeri* and *A*. *clavatus*) and it was found that more than two thirds of *A*. *fumigatus* biosynthetic gene clusters were absent [80]. The subtelomeric gene (*hrmA*) which regulates a cluster of genes that facilitate adaptation to very low oxygen conditions was present in *A. fumigatus* and absent in close-related non-toxigenic species [81]. In the second model, most genetic determinants of virulence were highly conserved in closely related species; for instance, in a genomic comparison of *A*. *fumigatus* with its close, non-toxigenic relative (*A*. *fischeri*), 48 of 49 known genetic determinants of virulence were highly conserved in *A*. *fischeri* [80]. This ended the long-held notion that differences in virulence among organisms is entirely due to their differences in gene content, which most genomic studies of fungal pathogens and non-pathogens have been based upon. It is, however, the case that genes associated with pathogenicity could be shared or absent amongst all pathogens, or be uniquely present or absent in each pathogen [79].

## 4. Virulence Factors of *Aspergillus* Species

*Aspergillus* species have developed a remarkable tolerance to highly stressful circumstances. Their ability to penetrate host defences, to take over the host, and high cell energy-generating capacity, among other attributes, have greatly contributed to their efficiency as opportunistic pathogens. The ability of *Aspergillus* species to thrive, succeed, and dominate in diverse ecological environments has been attributed primarily to some of the following features:Their conidia contain a rodlet layer in their surfaces, which binds covalently to the cell wall. This layer contributes to spore dispersion and fixation to the soil. It also helps to mask recognition of the conidia by the immune system, thereby preventing an immune response [82]. Furthermore, *Aspergillus* spores are hydrophobic and readily airborne, with the potential of germinating in a wide range of environmental conditions. These spores are among the microbial cells with the greatest longevity, surviving for 60 years or longer [46].Galactosaminogalactan (GAG) is a component of the *Aspergillus* cell wall that is expressed during conidial germination and hyphal growth. It induces the anti-inflammatory cytokine interleukin-1 receptor antagonist, making individuals more susceptible to aspergillosis [83]. Additionally, production of aerial hyphae enhances oxygen uptake, which is unique to members of the genus *Aspergillus*.*Aspergillus* species are nutritionally versatile in diverse environments including host tissues. Expression of multiple enzyme-linked genes that regulate metabolic pathways allows the fungi to be highly effective at upregulating the tricarboxylic acid cycle and to conveniently metabolize other secondary carbon sources [84,85].Secretion of a variety of proteases (degrading enzymes) by *Aspergillus* species enable the fungus to saprotrophically infect a wide variety of hosts [86,87]. For example, proteases with elastinolytic activity also function as virulence factors by degrading the structural barriers of the host and thereby facilitating the invasion of host tissues [88,89,90].Trace metal ions (iron and zinc) also contribute to virulence. Zinc is essential for a variety of biochemical processes in fungi, including the proper regulation of gene expression for cellular growth and development. A homeostatic relationship has been established between zinc and the virulence of *Aspergillus* species, as zinc transporters are required for growth within a host [80,91,92]. Also, iron is a necessary component of many biosynthetic pathways in fungi and is therefore required in pathogenesis. Since free iron is scarce in the human body, some *Aspergillus* species are able to transport and store ferric ions [93].Most *Aspergillus* spores are thermotolerant and their small and readily airborne asexual spores contribute greatly to their pathogenicity [94]. Most members of the genus *Aspergillus* have an optimum temperature range between 30 °C and 40 °C, with the ability to survive in temperatures as low as 12 °C and as high as 85 °C due to their *thtA* and *cgrA* genes. These are involved in their thermotolerance [94,95]. The ability of these species to survive in a wide range of water activity (optimal being 0.970, minimal at 0.770, and survivable at 0.640) should not be underestimated [20]. Under NaCl-induced stress, *Aspergillus* species are able to produce large amounts of cellulases, expediting the breakdown of cellulose that can be used for growth and energy generation [94]. Also, the ability to respond to multiple environmental stresses, including antifungal drugs, and the capacity to biosynthesize a range of structurally diverse secondary metabolites, are advantageous for the survival of this fungal group [3,80,96,97].

Figure 1 below summarises factors enhancing pathogenicity in *Aspergillus* species.

## 5. Control of Field *Aspergillus* Species

Many challenges faced by the agricultural sector include pests, diseases, and abiotic stresses. These drastically affect crop yield and threaten global food security. Several viral, fungal and bacterial plant pathogens are known to contribute to these losses. Fungi are adapted to different and extreme environmental conditions and, as a result, are able to colonize many agricultural products [98]. Fungal strains are the main herbicide-degrading microorganisms and are the most tolerant group of microorganisms to environmental stressors [99]. They secrete large amounts of extracellular enzymes that lead to enhanced xenobiotic biodegradation [100]. Even though fungi have colonized numerous crops, cereals are the most susceptible to these fungal secondary metabolites, with maize being the most affected (Pereira et al., 2014). The degree to which fungal growth and mycotoxin production will influence these crops depends on pre- and post-harvest factors [27,37]. Verheecke et al. [101] stated that soil and crop management practices remain the principal method of preventing pre-harvest mycotoxigenic fungal infestation and ensuing mycotoxin contamination. Miller [102] estimated a 40% loss in agricultural crops in developing countries because of mould infestation, with recent estimates ranging from 62–80% [7]. Economic losses linked with fungal secondary metabolite contamination are difficult to ascertain due to the complex nature of the food system, which varies annually. Nonetheless, some studies have indicated losses in the billions of dollars [29,103]. In the current state of global food insecurity, worsened by the COVID-19 pandemic, the importance of methods for the elimination and minimisation of crop losses, especially in the field, will become greater. Control of fungal contamination of crops in the field will go a long way to minimize post-harvest contamination of mycotoxins. Information on mycotoxin contamination and its toxicological consequences in the soil ecosystem on a regional basis is limited. Understanding pre-and postharvest mycotoxin risks, and whether new or pathogenic *Aspergillus* species could emerge due to climate change, is crucial for future mycotoxin mitigation efforts [59]. The World Soil Charter recognizes the relevance of soil biodiversity in supporting healthy soil functions which provide, regulate, and maintain various ecosystem services. Protecting soil biodiversity is critical for the success of the United Nations Decade on Ecosystem Restoration (2021–2030) [57]. A balance needs to be struck between supporting a healthy soil biodiversity and the management of crop toxigenic species in order to ensure food security.

The fate and consequences of aflatoxin (AF) contamination in soil and on other soil organisms remains unclear and might be a high risk to soil health [23]. Food crops cultivated in AF-contaminated soil absorb AFs into their leaf, stem, and root tissues [104], leading to reduction in crop yield and resulting in food shortages. These contaminated food crops are hardly discarded, especially in most rural communities in sub-Saharan Africa, but end up in the animal/human food chain, affecting the health of these organism/individuals [105]. This form of AF loading into the soil increases natural systemic contamination and alters the soil’s ecological balance and could change its physicochemical properties and biotic parameters. When mycotoxin-contaminated residues are left in the field to decompose naturally, mycotoxins have the ability to seep and contaminate groundwater, depending on the soil type [106]. This negatively affects the soil quality, as there will be an increase in mycotoxin systemic contamination and the soil’s ecological balance will be altered. Soil quality is of the utmost importance, as it affects food production systems. Regulatory laws on proper handling, managing, and disposal of AF-contaminated residues are unfortunately scarce, with options for the removal of AF-contaminated residues being restricted to incineration or working the contaminated crops or feed back into the soil [107]. The question is, are these methods environmentally sustainable?

Numerous antimycotic compounds to control the spread of these fungal pathogens exist. Azoles, for example, are unsaturated aromatic molecules that have been widely used as antifungals. The triazoles have been used as antifungals in agriculture due to their systemic distribution in treated plants, high efficiency, and their broad spectrum of target pathogens [108]. Human resistance to antibiotics is increasing and has been assumed to arise, among others, from azole fungicides use in agriculture [108]. Azole fungicide resistance has been found in environmental isolates in China [109,110,111]. It is interesting to note that, while some authors have reported environmental resistance frequencies of up to 20%, others have found no incidence of environmental resistance [112,113]. Barber et al. [108] found no proof that azole fungicide use in crops significantly contribute to resistance in *A. fumigatus*. From this literature review, the environmental resistance rates reported have been substantial. It is worth noting that the emergence of resistance is not systematically obtained after antifungal drug exposure, as revealed by Kano et al. [114], who sprayed fields twice a year with tetraconazole. Ren et al. [110] pointed out that the acquisition of mutations in the azole target by environmental strains led to cross-resistance between azole antifungals in the environment and the clinic. Hence, fungal exposure to azole compounds in the environment resulted in cross-resistance to medicinal triazoles. Also, sexual reproduction facilitates the emergence of azole resistance. Compost heaps containing residual azole fungicide are warm, dark environments, low in oxygen and high in carbon dioxide, that promote sexual reproduction and consequent genetic recombination [111]. The intention to solve the problem of fungal pathogen contamination of crops in the field using azoles should supersede the resistance problem created in the process.

Although numerous strategies have been applied globally to reduce pre-harvest fungal contamination, biological control methods so far remain the most reliable and environmentally sustainable solution to the control of soil toxigenic fungi [103,115,116,117,118,119,120] (Table 1). For instance, selected isolates of *Trichoderma* species have been shown to be efficient suppressors of soil-borne pathogens through the mechanisms of competition, antibiosis, and mycoparasitism [115]. Also, the efficiency of biocontrol strategies has been supported. For example, it was found that atoxigenic strains of *A. flavus* isolates applied before harvest out-compete their toxigenic counterparts. This strategy has been applied and validated in the U.S., Europe, and some parts of Africa [103,116]. A reduction of between 80% to 95% of toxigenic *A. flavus* has been achieved in corn fields in the USA and Italy using bioplastic granules carrying an atoxigenic strain of *A. flavus* [116]. The main benefit of this control method is that the effects of the treatment can last relatively long and the debris associated with treated crops, fields, and other colonized organic matter may extend the benefits of atoxigenics well beyond the year of treatments [117,118]. Biocontrol carryover influences the composition of *A. flavus* populations from the previous season into the next. The movement of atoxigenics, instead of aflatoxin producers, with the treated crop throughout the value chain [119] constitutes an advantage. Another successful and safe biocontrol method has recently been applied in maize fields in Egypt, where endophytic fungi (*A. fumigatus*) were used to reduce the growth of their aflatoxigenic endophytic counterparts (*A. flavus*). *A. fumigatus* inhibited the growth of *A. flavus* by 77% and further reduced the reduced aflatoxin production by 90.9% [120].

## 6. Conclusions

The biodiversity of five members of *Aspergillus* sections known to cause diseases was explored in this review in terms of ecological dominance, pathogenicity potential, and control techniques. The ability of *Aspergillus* species to adapt to different environments is partly due to their structural constituents like the rodlet layer in their surfaces which binds covalently to the cell wall, functions in spore dispersion and fixation to the soil, and helps masks recognition of the conidia by the immune system [82]. Also, GAG in their cell wall, expressed during conidial germination and hyphal growth, induces the anti-inflammatory response, making individuals more susceptible to aspergillosis [83]. In addition to their structural advantages, the production of aerial hyphae enhances oxygen uptake, which is unique to members of the genus *Aspergillus*. Apart from structural advantages, *Aspergillus* species are nutritionally versatile in diverse environments, including in their metabolic adjustment to nutritional and biophysical challenges and host tissues. Their ability to secrete various degrading enzymes enables the fungus to infect a wide variety of hosts saprotrophically [86,87]. Also, their high efficiency in responding to other significant environmental stressors confers an advantage on this group of fungi [3,80,96,139]. Lastly, the remarkable discovery of sexual stages among members of the genus *Aspergillus* that were formerly assumed to be asexual confers an ecological advantage and increased pathogenicity within this group.

Structural, nutritional and physiochemical advantages posed by members of the genus *Aspergillus* constitute an environmental advantage in the face of climate change. By implication, the emergence of new disease-causing *Aspergillus* species and mycotoxin contamination risks in soil ecosystems will continue to increase in the tropical and subtropical regions. Hence, the extent of the contamination risk will vary from one geographic region to the other depending on local climatic conditions. In the polar regions of the world, the climate change effects will be minimal. Battilani et al. [14] demonstrated in a climate model that, if global warming continues as a result of climate change in the tropics with an already hot climate, toxigenic *Aspergillus* species will disappear; an unexpected benefit of global warming. But then, the World Soil Charter recognizes the relevance of soil biodiversity in supporting healthy soil functions, and a balance needs to be struck between supporting healthy soil biodiversity and the management of crop toxigenic species to ensure food security. Farming techniques such as irrigation, intercropping and crop rotation are good practices that should be applied, while planting of contaminated seed, late planting and monoculture should be discouraged, as these good techniques have shown that, if practiced properly, they reduce the infestation of the soil and subsequently the crops by toxigenic fungi and will go a long way to avoiding/reducing mycotoxin contamination of crops [105]. Finally, in mycotoxin endemic regions, reactions such as the complete removal of aflatoxin-contaminated residues are recommended. Techniques such as incineration and burying are the main methods, as recommended by the World Soil Charter Organization.

Amidst the numerous strategies that have been applied globally to reduce pre-harvest mycotoxin contamination, biological control methods so far remain the most promising and environmentally sustainable solution to the control of toxigenic fungi in the field. Novel biological control agents need to be explored, with *Trichoderma* and *Rhizobia* species showing potential. We recommend more research work be done on finding possible non-toxigenic *Aspergillus* species on their toxigenic counterparts isolated from the same field (i.e., endophytes), rather than importing new strains or isolates which might be costly; moreover, problems of adaptability to local climate might reduce and/or inhibit efficiency.

## Figures and Tables

**Figure 1 jof-09-00766-f001:**
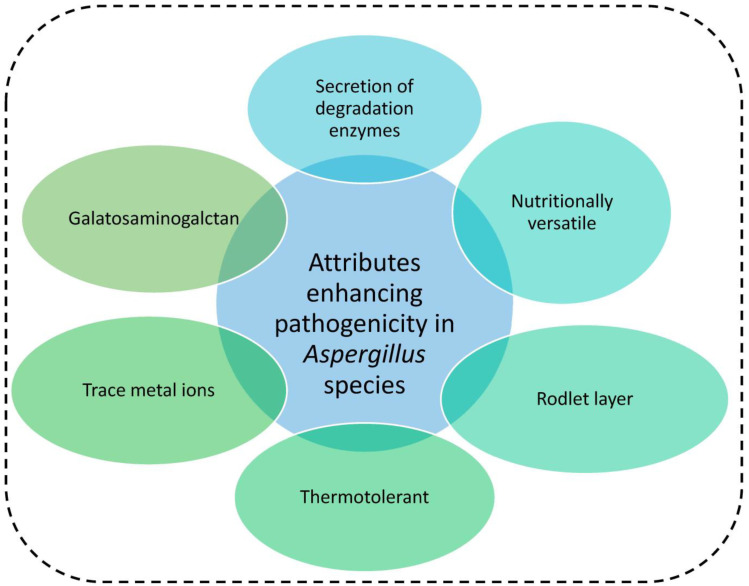
Factors enhancing pathogenicity in *Aspergillus* species.

**Table 1 jof-09-00766-t001:** Biological, chemical and physical control methods of field *Aspergillus* species.

Method	Principle	Merits	Demerits	References
Use of azoles	Antifungal	High efficiency, broad spectrum of target pathogens	Developed both clinical and environmental resistance	[108,109,110,111]
Aluminosilicates	Binding ability, are produced synthetically or extracted from clay mines.	Mycotoxin binders	Ambiguous methodologies used for evaluation, contrasting results	[121]
Benlate (Methyl-1-Butyl-Carbonomyl-1-2-benzimidizole carbonate)	Systemic fungicide against important fungal pathogens	Relatively safe, broad spectrum against pathogens	Not easily accessible	[122]
Calcium or lime application	Their use on farm yard manure and cereal crop residues as soil amendments have shown to be effective in reducing *A. flavus* contamination	Thickens the cell wall and accelerates pod filling, while manure facilitates growth of micro-organisms that suppress soil infections	N/D	[123]
Appropriate use of fertilizers; insecticide and herbicides		Raise crop yield	Pose environmental risks	[123,124,125]
Residual management	Incineration/bury	The fungus systemic circle is broken	Contamination of underground water. Incineration is not environmentally sustainable	[107,123,126]
Proper crop rotation	Crop rotation models have shown low levels of mycotoxins as compared with crops from monocropping systems	The fungus infectious cycle is disrupted. Improves soil fertility	Challenge in finding the right crop rotation combination; for instance, maize/wheat is an inappropriate combination, as both crops have been proved to be susceptible to fungal infection	[61,62,127]
Appropriate cultivar, early sowing and harvest dates	Drought-resistant, early maturing cultivars are important. Changing of planting/harvesting dates in response to unpredictable onset of rains is key	Reduces drought stress. Minimizes crop exposure to drought, rewetting and others	N/A	[121,123,125,128,129]
Irrigation	Artificial application of controlled amounts of water to land to assist in crop production	Reduces drought Stress and encourages fungal growth. Boost crop production	Irrigation schemes are expensive to acquire	[130]
*Trichoderma* species	Ability to produce both volatile and non-volatile metabolites that adversely affect growth of different fungi.	Considered a more natural and environmentally acceptable control method	N/D	[115,122]
*Aspergillus* species	Competition	Long-term treatment effects due to carry over into the next season	N/A	[103,116,117,118,119,131]
White rot fungi (*Phanerochaete sordida*, *Armillariella tabescens*, *Pleurotus ostreatus*, *Peniophora* sp.)	Degrading abilities of a broad spectrum of structurally diverse toxic environmental pollutants	No residual toxicity observed in some products	Toxicity of most degradation products not determined	[132,133,134,135]
*Aspergillus fumigatus* *A. fumigatus*	Anti-fungal Growth inhibition	Relatively safe and highly efficient	No known	[120]
Phytochemicals of plant extracts such as polyphenols, polyenes and essential oils	Fungicidal	Highly effective and environmentally sustainable.	Low stability and solubility and high cost	[136]
Fungal concoction (*Monascus* species and *T. harzianum*)	Fungal growth inhibition	Environmentally friendly	No known	[137]
*Streptomyces philanthi* strain RL-1-178	Fungal growth inhibition	Highly efficient (85–100%)	No known	[138]

N/D = Not determined, N/A = Not applicable.

## Data Availability

Not applicable.

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
