# Peer review of "Soil Aspergillus Species, Pathogenicity and Control Perspectives"

_jof, 2023, doi:10.3390/jof9070766_

Round 1

Reviewer 1 Report

The manuscript entitled “Biodiversity of Aspergillus species, their virulence factors and control perspectives” tries to identify the biodiversity of representative species from five Aspergillus sections known to cause diseases in humans and explore their ecological dominance, pathogenicity potential, and possible control techniques. However, this quality of work is not enough to publish in Journal in Fungi. My decision is that the manuscript is unsuitable for publication as it stands.

1.    The quality of the review heavily relies on the graphics, but the charts and tables do not reflect any summaries or overviews of the research of this field.

2.    There are many incorrect language expressions in the article, such as the first sentence of the abstract.

3.    The manuscript does not adequately summarise the ecological attributes that contribute to the pathogenicity of certain Aspergillus genus members.

4.    The biological control techniques recommended are not highlighted.

5.    The article raises questions about balancing healthy soil biodiversity and toxic fungi species control. However, there is no exploration of these measures.

6.    The language structure and grammar used in the article need significant improvement.

7.    There are lines covering the text in Figure 1, and the bullets in the rightmost column are not aligned.

8.    The first letter of the word “artificial” in the penultimate row of Table 1 is not capitalized.

9.    Line 51, the details of information about metabolites should be added.

10.  Line 184, please list the exact pathogenicity of different Aspergillus species to humans. 

11.  The first letter of the word “factors” in the Figure 2 is not capitalized.

12.  Line 375, the references of strategies that have been applied globally to reduce pre-harvest fungal contamination were not cited.

The language structure and grammar used in the article need significant improvement.

Author Response

Reviewer 1

The manuscript entitled “Biodiversity of Aspergillus species, their virulence factors and control perspectives” tries to identify the biodiversity of representative species from five Aspergillus sections known to cause diseases in humans and explore their ecological dominance, pathogenicity potential, and possible control techniques. However, this quality of work is not enough to publish in Journal in Fungi. My decision is that the manuscript is unsuitable for publication as it stands.

  1. The quality of the review heavily relies on the graphics, but the charts and tables do not reflect any summaries or overviews of the research of this field.

  • The title has been redesign to reflect more on the content and hence, charts and tables.

  1. There are many incorrect language expressions in the article, such as the first sentence of the abstract.

  • The sentence has been rephrased. Thank you for that observation

  1. The manuscript does not adequately summarise the ecological attributes that contribute to the pathogenicity of certain Aspergillus genus members.

  • The inadequacy of information on the ecological attributes that contribute to the pathogenicity of certain Aspergillus species is because Literature is scarce or limited in other Aspergillus species while others have been extensively covered might be due to their socio-economic relevance predominantly from the perspectives of human and livestock health.

  1. The biological control techniques recommended are not highlighted.

  • Table 1; even though is a summary of all the control methods of field Aspergillus species, biological methods have been listed among. For example; the use of Trichoderma species, white rot fungi, fumigatus and other Aspergillus species have successfully been applied in the USA and also in Africa.

  1. The article raises questions about balancing healthy soil biodiversity and toxic fungi species control. However, there is no exploration of these measures.
  • Still in table 1, physical control methods of field Aspergillus species such as irrigation, farming system (appropriate crop rotation methods), and incineration and or burying of debris of previous farming seasoned as control methods that will provide a balance between healthy soil biodiversity and toxic fungi species. In all, regulatory laws on proper handling, managing, and disposal of AF-contaminated residues are unfortunately scarce, with options for the removal of AF-contaminated residues being restricted to incineration or working the contaminated crops or feed back into the soil according to East Africa community on the regulation.
  1. The language structure and grammar used in the article need significant improvement.
  • The services of language editing were employed.
  1. There are lines covering the text in Figure 1, and the bullets in the rightmost column are not aligned.
  • This has been adjusted accordingly.
  1. The first letter of the word “artificial” in the penultimate row of Table 1 is not capitalized.
  • It has now been capitalized.
  1. Line 51,the details of information about metabolites should be added.
  • such as; aflatoxins, gliotoxin, patulin, cyclopiazonic acid, and ochratoxin
  1. Line 184,please list the exact pathogenicity of different Aspergillus species to humans. 
  2. The first letter of the word “factors” in the Figure 2 is not capitalized.
  • It has now been capitalized.
  1. Line 375, the references of strategies that have been applied globally to reduce pre-harvest fungal contamination were not cited. (line384)
  • They have now been cited; Weaver & Abbas, 2019, Boughalleb-M’Hamdi et al., 2018, Mauro et al., 2018, Cotty, 2006, Cotty et al., 2007, Atehnkeng et al., 2014, Abdelaziz et al., 2022

Reviewer 2 Report

The review discusses the genus Aspergillus, that include agricultural pests and producers of various metabolites, posing a threat to global food safety. The review explores the ecological factors that have contributed to the success of pathogenicity in certain Aspergillus species. The text also covers the virulence factors, ecology, and potential biological control techniques for managing these fungi.

I believe that the review has value to the literature, and fits the scope of JoF, however I have some questions.

-the title seems inappropriate to me, as it is not a review of biodiversity. The work is mainly focused on field and soil aspects, even the virulence and pathogenicity part is linked to these aspects. Please think of a more representative title.

-In lines 49-52, the authors use the expression: "these cryptic species", which cryptic species are the authors referring to?

It seems that they treat the term "cryptic species" in a generalized way, how do the authors define a cryptic species?

-There also seems to be here, and throughout the text, a confusion of what constitutes a pathogenic species and a toxigenic species, they are different things.

-What do the authors mean by: " Soil is the natural home for A. flavus and black aspergilli, giving it the environmental advantage of mycotoxin production [9]",Why would soil provide any environmental advantage for mycotoxin production?

The reference provided does not support this claim.

-in the stretch: "The impacts of climate change on the emergence of new criptic Aspergillus species and mycotoxin production are uncertain and if climate change continues  at its current pace, mycotoxin contamination risks in soil ecosystems will certainly in-crease [10]. " The authors state that the impacts of climate change are uncertain, however, in the following lines they state that they will certainly increase the risks of contamination by mycotoxins?

Again, there seems to be some confusion with the term cryptic species, I believe that the authors' understanding is wrong.

-about the term: "mycotoxin contamination risks in soil ecosystems", Could you clarify what the impacts of contamination by mycotoxins in soil would be? The problem with mycotoxins is food contamination. 

-Figures 1 and 2 are uninformative and graphically ugly. Improve or remove them.

-In the conclusion section, the authors are too repetitive, please write more succinctly.

Minor editing of English language required

Author Response

Reviewer 2

The review discusses the genus Aspergillus, that include agricultural pests and producers of various metabolites, posing a threat to global food safety. The review explores the ecological factors that have contributed to the success of pathogenicity in certain Aspergillus species. The text also covers the virulence factors, ecology, and potential biological control techniques for managing these fungi.

I believe that the review has value to the literature, and fits the scope of JoF, however I have some questions.

-the title seems inappropriate to me, as it is not a review of biodiversity. The work is mainly focused on field and soil aspects, even the virulence and pathogenicity part is linked to these aspects. Please think of a more representative title.

  • The title has been changed to; Soil Aspergillus species, pathogenicity and control perspectives

-In lines 49-52, the authors use the expression: "these cryptic species", which cryptic species are the authors referring to?

  • Any species that are physically or morphologically indifferent from one another but one is of clinical concern, example is flavus and A. oryzae. If A. is among a population of which majority is A. oryzae, it might be mistaken for A. oryzae, in such a scenario A. flavus can be refer to as the cryptic species.

 It seems that they treat the term "cryptic species" in a generalized way, how do the authors define a cryptic species?

  • Species that are physically or morphologically indifferent from one another but one is pathogenic depending on the prevailing environmental conditions and one can be mistaken for the other.

-There also seems to be here, and throughout the text, a confusion of what constitutes a pathogenic species and a toxigenic species, they are different things.

  • Indeed they are, the confusion has been sorted out. Thank you.

-What do the authors mean by: " Soil is the natural home for A. flavus and black aspergilli, giving it the environmental advantage of mycotoxin production [9]",Why would soil provide any environmental advantage for mycotoxin production?

  • Soil is the natural habitat for toxin-producing fungi of Aspergillus species example flavus and black aspergilli, giving it the environmental advantage of mycotoxin production (Winter and Pereg, 2019).

The reference provided does not support this claim.

  • The reference has been corrected. Thank you

 -in the stretch: "The impacts of climate change on the emergence of new criptic Aspergillus species and mycotoxin production are uncertain and if climate change continues  at its current pace, mycotoxin contamination risks in soil ecosystems will certainly in-crease [10]. " The authors state that the impacts of climate change are uncertain, however, in the following lines they state that they will certainly increase the risks of contamination by mycotoxins?

  • The impacts of climate change on the emergence of new Aspergillus species and mycotoxin production are uncertain and if climate change continues at its current pace, mycotoxin contamination risks in soil ecosystems will increase or decrease depending on the region under consideration (Batillani et al., 2016, Sibakwe et al., 2017).

Again, there seems to be some confusion with the term cryptic species, I believe that the authors' understanding is wrong.

  • The sentence has been paraphrase without the word ‘cryptic’.

 -about the term: "mycotoxin contamination risks in soil ecosystems", Could you clarify what the impacts of contamination by mycotoxins in soil would be? The problem with mycotoxins is food contamination. 

  • Agreed, food cultivated in such soil risk contamination by these mycotoxins.

-Figures 1 and 2 are uninformative and graphically ugly. Improve or remove them.

  • Figure 1 is removed.

-In the conclusion section, the authors are too repetitive, please write more succinctly.

  • Noted with thanks.

Round 2

Reviewer 1 Report

1.    The sentence “Figure 1 below summarises factors enhancing pathogenicity in Aspergillus species” lacks a colon and capitalized first letter.

2.     Insufficient contextualization: The conclusion briefly mentions climate change, emerging disease-causing species, and mycotoxin contamination risks without providing further context or discussing their implications. More information is needed to fully understand the relevance of these factors.

3.    Lack of context and clear background information in the introduction: While the importance and research significance of the Aspergillus genus are mentioned, it is not explicitly discussed in which areas or aspects this genus has influence, and there is a lack of relevant background information.

4.    The references cited in this conclusion are somewhat dated. It is advisable to include more recent literature within the past five years. This will ensure that the information presented remains up-to-date and accurate, better supporting the conclusion and reflecting the latest developments in the field.

5.    To improve this conclusion, it should summarize the main findings and emphasize the significance of the discussed factors in relation to biodiversity, pathogenicity, and control techniques of Aspergillus species.

6.    Lack of specific recommendations: While biological control methods are mentioned as a sustainable solution, the conclusion does not provide specific recommendations or suggestions for further research or action, it should provide specific recommendations or directions for future research, addressing the challenges posed by climate change, emerging species, and mycotoxin contamination risks. A more cohesive and comprehensive ending will help provide closure to the entire review.

7.    The conclusion does not effectively summarize the main points discussed in the preceding text and lacks a clear connection to the overall topic of biodiversity, pathogenicity potential, and control techniques of Aspergillus species.

Grammar and wording issues: Some sentence structures are unclear, and the choice of words is imprecise or unclear, requiring revisions and modifications to enhance readability.

Author Response

1.    The sentence “Figure 1 below summarises factors enhancing pathogenicity in Aspergillus species” lacks a colon and capitalized first letter.
Appropriate corrections have been made. Thank you
2.    Insufficient contextualization: The conclusion briefly mentions climate change, emerging disease-causing species, and mycotoxin contamination risks without providing further context or discussing their implications. More information is needed to fully understand the relevance of these factors.
This has been address properly (see manuscript highlighted in yellow under conclusion). 

3.    Lack of context and clear background information in the introduction: While the importance and research significance of the Aspergillus genus are mentioned, it is not explicitly discussed in which areas or aspects this genus has influence, and there is a lack of relevant background information.
Vital background information has been added to the introduction. (see manuscript highlighted in yellow in the introduction section). 

4.    The references cited in this conclusion are somewhat dated. It is advisable to include more recent literature within the past five years. This will ensure that the information presented remains up-to-date and accurate, better supporting the conclusion and reflecting the latest developments in the field.
Up to date references have been added where necessary for example; Eskola et al., 2020, Human, 2018, Qi et al., 2023, Mwatabu et al., 2023, Boukaew et al., 2023.

5.    To improve this conclusion, it should summarize the main findings and emphasize the significance of the discussed factors in relation to biodiversity, pathogenicity, and control techniques of Aspergillus species.
This has been address properly (see manuscript highlighted in yellow). 

6.    Lack of specific recommendations: While biological control methods are mentioned as a sustainable solution, the conclusion does not provide specific recommendations or suggestions for further research or action, it should provide specific recommendations or directions for future research, addressing the challenges posed by climate change, emerging species, and mycotoxin contamination risks. A more cohesive and comprehensive ending will help provide closure to the entire review.
Proactive recommendations have been added such good farming techniques should be adopted while other farming techniques that have proven to favour the growth of toxigenic fungi discouraged. While in mycotoxin endemic regions, reactions such as complete removal of aflatoxin-contaminated residues are recommonded. Finally, we recommend more research work be done on finding possible non toxigenic Aspergillus species on their toxigenic counterparts isolated from the same field (i.e endophytes), rather than importing new strains or isolates which might not only be costly but problems of adaptability to local climate might reduce and/or inhibit efficiency. (Emphasis is placed on endophytes).

7.    The conclusion does not effectively summarize the main points discussed in the preceding text and lacks a clear connection to the overall topic of biodiversity, pathogenicity potential, and control techniques of Aspergillus species.

This too has been addressed properly to the best of my ability. Thank you

Round 3

Reviewer 1 Report

It is can be accepted.